# A Spiral Single-Layer Reticulated Shell Structure: Imperfection and Damage Tolerance Analysis and Stability Capacity Formulation for Conceptual Design

**Huijuan Liu [1], Fukun Li [2,*], Hao Yuan [3,*], Desheng Ai [4] and Chunli Xu [5]**

[1] Key Laboratory of Disaster Prevention and Structural Safety of Ministry of Education, Guangxi Key Laboratory of Disaster Prevention and Engineering Safety, Guangxi Laboratory on the Study of Coral Reefs in the South China Sea, College of Civil Engineering and Architecture, Guangxi University, Nanning 530004, China; 20110057@gxu.edu.cn

[2] College of Civil Engineering and Architecture, Guangxi University, Nanning 530004, China

[3] SDR Engineering Consultants, Inc., Tallahassee, FL 32308, USA

[4] Shanghai Zhongxin Architectural Design and Research Institute Corp., Ltd., Shanghai 200083, China; aidesheng222@163.com

[5] Shanghai Baoye Group Corp., Ltd., Shanghai 200941, China; jx2020@vip.163.com

[*] Correspondence: gxussgmsc@163.com (F.L.); hao.yuan@uconn.edu (H.Y.)

**Abstract:** Single-layer reticulated shell structures are widely used, but their stability performance is not ideal. Moreover, they are sensitive to structural damage and imperfections, while the existing conventional design methods of increasing the cross-section, strengthening corrosion protection, and densifying the structural grid are not economical. This study employs a modified and bionic structure—a spiral single-layer reticulated shell structure—to solve the problem. First of all, according to the current Chinese design codes, its mathematical model and geometric model are designed. Then, its damage and imperfection tolerances are analyzed and compared with a traditional single-layer reticulated shell. We then propose a universal bearing capacity formula. Our research conclusions prove that the spiral single-layer reticulated shell structure has a higher tolerance to damage and imperfections while maintaining stability. Moreover, the precise bearing capacity formula proposed will help engineers to efficiently select the structure configurations in the conceptual design phase. Therefore, the spiral single-layer reticulated shell structure is worthy of popularization and application in engineering practice.

**Keywords:** structural stability; high imperfection/damage tolerance; spiral single-layer reticulated shell; structural design; structural analysis; characterization of bearing capacity

## 1. Introduction

Single-layer reticulated shells have been widely used in structures for their light weight, aesthetics, and ability to span large space [1]. Based on the configuration of reticulation, traditional single-layer reticulated shells can be classified as radial ribbed, Schwedler, Lamella, Kiewitt, geodesic, three-way latticed, two-way latticed, etc., ref. [2]. However, these traditional single-layer reticulated shells have significant stability issues under external loads (including wind loads), compared with other common structural types. The stability performance of single-layer reticulated shells was found to be closely related to structural imperfection and damage. Specifically, the geometric imperfection of nodes [3] and the corrosion damage of members [4] can significantly reduce the structural bearing capacity [5]. To design such structures to be sufficiently safe, engineers often have to strictly control installation [6] and use a higher anti-corrosion level [7]. In some scenarios, higher strength materials, refined reticulation, and even anti-corrosion equipment [8] have to be used. However, these measures are never perfect ways to secure the structural stability since they consume more materials, raise the cost, and even affect the aesthetic appearance.

To fundamentally overcome the shortcomings of poor stability and imperfection/damage tolerance among single-layer reticulated shells, a novel structural configuration could be an ideal solution. Such a configuration needs to achieve adequate stability without greater costs, compared with traditional configurations.

After some investigation, we found that the spider web found in nature [9] is a good source of ideas for a reticulated shell configuration. The spider web appears to have a spiral configuration, which is a structural configuration containing the meaning of life [10]. The spiral configuration is not only unique and beautiful, but also has favorable properties bestowed by nature [11], such as versatile opening and growability. Spiral-inspired structures have an elegant appearance and a good mechanical performance at the same time [12], and there are multiple examples in the real world [13].

Therefore, this article establishes a modified type of spiral single-layer reticulated shell structure based on the idea of bionics, as well as the spiral equation in nature and the definition of the reticulated shell structure. Then, engineering design and structural analysis of the structure are performed and summarized. The structure is also studied for its tolerance to node imperfections and component damage, by comparing with a traditional radial ribbed reticulated shell. Finally, the modified structure's stable bearing capacity is formulated to demonstrate its characteristics and practicability, and can further be used as a conceptual design tool. When carrying out the stability analysis for the final formulation study, this study adopts a new imperfection analysis method with higher efficacy. The specific algorithm is introduced in Section 5.

## 2. A Modified Structure Design of Single-Layer Reticulated Shell Structure

### 2.1. Mathematical Model of the Modified Structure

Common three-dimensional spirals mainly include Archimedes spirals, Fermat spirals, equiangular spirals, and others. Archimedes spirals are the most common in nature. Additionally, considering the convenience of construction, for the modified structure's configuration, we chose the Archimedes spiral to build a three-dimensional model (Figure 1). We used the cylindrical coordinate system to build the model. Its mathematical equation in the cylindrical coordinate system $(r, \theta, z)$ is expressed as follows:

$$r = a + b\theta \tag{1}$$

$$z = \sqrt{R^2 - r^2} - H \tag{2}$$

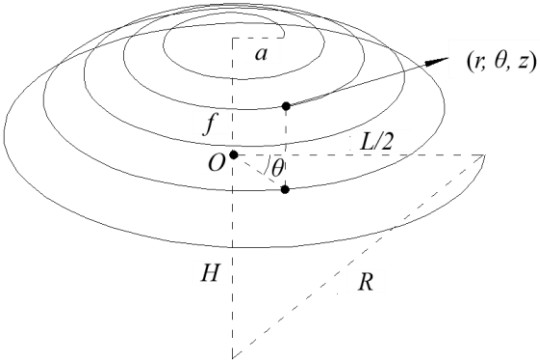

**Figure 1.** Archimedes spiral.

In the above formulas, $a$ and $b$ are constants for a particular curve, $a$ represents the distance from the starting point of the spiral to the instantaneous origin of polar coordinates and $b$ is the value of $r$ corresponding to the increase in each unit angle of the spiral. $z$ is the vertical coordinate of each node. We have the spiral ends at the maximum radius ($r = r_{\max} = L/2$). The $z$ axis starts at $O$ pointing upward. As shown in Figure 1, $L$ and $f$ are the horizontal span and vertical height (rise) of the reticulated shell structure, respectively.

The rise-to-span ratio $r_s = f/L$. $1/R$ is the curvature of the spherical surface where the spiral is located and $R = \sqrt{r_{max}^2 + H^2}$. $a$ and $b$ can be determined in the design process to adapt the $r_{max}$ and the number of loops. Generally, full loops are assumed.

### 2.2. The Geometric Model of the Modified Structure

Using the mathematical model of the Archimedes spiral shown in Figure 1, we designed the geometric model of the spiral single-layer reticulated shell by the design specification of the space reticulated shell [14]. The corresponding model parameters with different rise-to-span ratio ($r_s$) are shown in Table 1. Additionally, the ring division frequency is 48 and the number of loops is 5. A spiral single-layer reticulated shell configuration is then constructed based on this. Here, $a$ = 3 m, $b$ = 1.2/$\pi$ m/rad. The plan view and elevation view of one model with $r_s$ = 1/6 are shown in Figure 2.

**Table 1.** Model parameters of the modified structure.

| $r_s$ | Maximum Radius $r_{max}$ (m) | Height/Rise $f$ or $Z$ (m) |
|---|---|---|
| 1/5 | 15 | 6 |
| 1/6 | 15 | 5 |
| 1/7 | 15 | 30/7 |

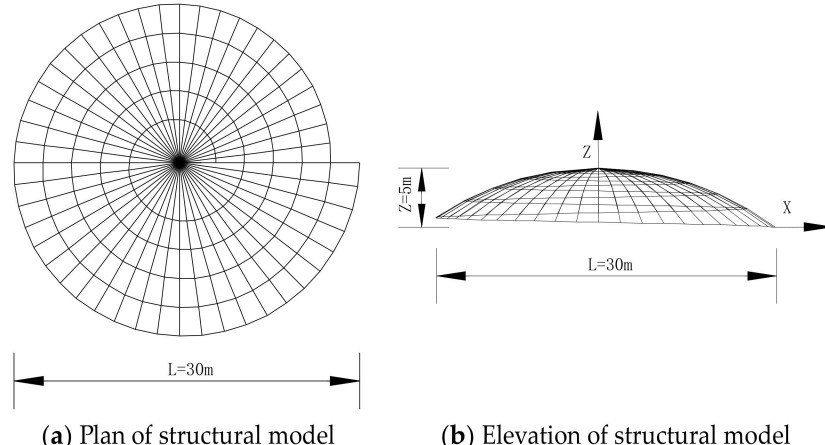

(**a**) Plan of structural model      (**b**) Elevation of structural model

**Figure 2.** Plan and elevation of the modified structure ($r_s$ = 1/6).

For comparative analysis and research, using the same span (to be exact, the same maximum radius $r_{max}$ = $L/2$, ignoring the slight span difference due to the growth feature of the spiral) and height/rise ($f$ or $Z$)—leading to the same $r_s$, as well as the same ring segmentation and loop number as the above model of the modified structure—we establish a corresponding radial ribbed single-layer reticulated shell model as a comparison model. The plan view and elevation view of the comparison model with $r_s$ = 1/6 are shown in Figure 3.

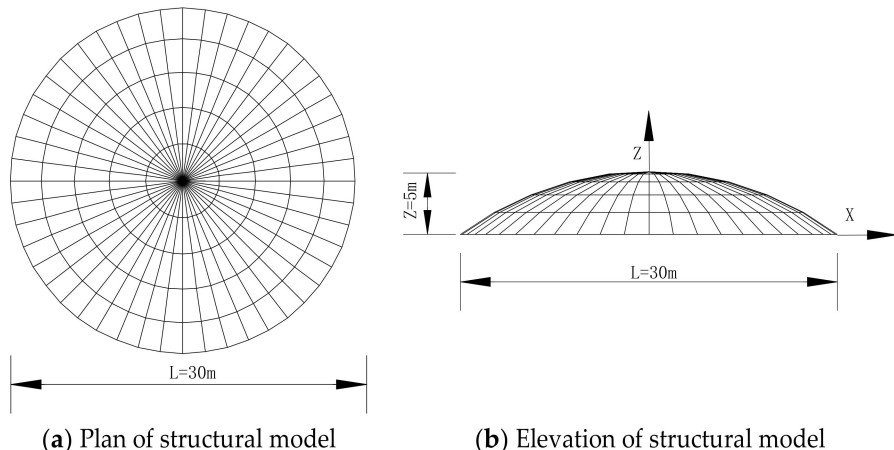

(**a**) Plan of structural model   (**b**) Elevation of structural model

**Figure 3.** Plan and elevation of the comparison structure ($r_s$ = 1/6).

### 3. Engineering Design of the Modified Structure

#### 3.1. Finite Element Models

According to their geometric parameters and the principles shown in Section 2.2, the corresponding finite element analysis models are built. Geometric nonlinearity is considered in the analysis. The material nonlinearity is not considered since the reticulated shell in this study buckles in the elastic regime. The steel used for the reticulated shell structure is assumed to be an ideal elastic material (isotropic), of which the elastic modulus *E* is 201 GPa, the shear modulus is 80.7 GPa, the Poisson's ratio is 0.3, and the mass density is 7850 kg/m$^3$. The section of the beam element is a hollow round tube of $\varphi$219 × 6 mm. It is assumed that the rods of the reticulated shell are fixedly connected, and the nodes at the periphery have three-dimensional hinge supports. These two structures are modeled using Ansys software [15]. The discrete element of the rod is BEAM188. The SURF154 element is used to transfer the uniform load of the structure from the surface to the nodes.

The virtual location of this structural project is Nanning, Guangxi, China. By the *GB50009-2012 Load code for the design of building structures* [16], when designing the modified structure and the comparison structure, the primary considerations are the three effects of the dead load, live load, and wind load, and their six load combinations [17]. The loads other than member self-weight are applied here as surface loads on the external envelope surface (the sphere on which the elements lie) of the reticulated shells in the analysis. The load cases and corresponding factor values of the three load types are shown in Table 2.

**Table 2.** Different load cases of the structure.

| Load Case | 1 | 2 | 3 | 4 | 5 | 6 |
|---|---|---|---|---|---|---|
| $r_G$ | 1.35 | 1.3 | 1.3 | 1.3 | 1.3 | 1.3 |
| $r_Q$ | 1.5 × 0.7 | 1.5 | 1.5 | 1.5 | 1.5 | 1.5 |

Note: The live load of Load Case 1 and 2 is roof live load. The live loads of Load Cases 3, 4, 5, and 6 are wind loads with directions of 0°, 90°, 180°, and 270°, respectively. $r_G$ denotes partial factor of dead load, $r_Q$ denotes partial factor of live load.

We noticed that the modified structure—the spiral reticulated shell—is an irregular structure, so the effect of different wind direction angles needs to be considered when wind loads are applied. On the contrary, the comparison structure—the ribbed ring reticulated shell—does not need consideration of the wind angle, and the wind direction of 0° is used for all calculations, as shown in Figure 4.

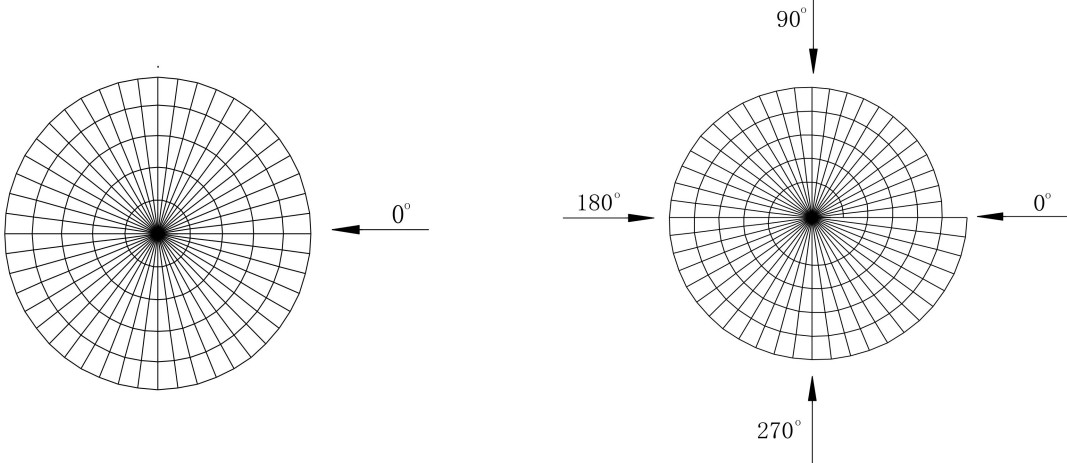

(**a**) Direction of wind load on the comparison structure     (**b**) Direction of wind load on the modified structure

**Figure 4.** The direction of wind load on the comparison structure and the modified structure.

### 3.2. Design Calculation of the Modified Structure

### 3.2.1. Support Reactions

The maximum support reactions of the modified structure and the comparison structure under different load cases are shown in Figure 5. $F_x$, $F_y$, and $F_z$ are the reaction force in the X, Y, and Z direction (refer to Figures 2b and 3b for the coordinate system following the right-hand rule). For the modified structure, the overall maximum $F_x$ is happening at Load Case 5 with the wind in the +X direction; the overall maximum $F_y$ is happening at Load Case 6 with the wind in the +Y direction; the overall maximum $F_z$ is happening at Load Case 1 with dead load and live load in control. For the comparison structure, Load Cases 3–6 are the same since the wind is fixed in the −X direction due to symmetry. For the same reason, $F_x$ and $F_y$ are equivalent for design. $F_y$ in Figure 5 is actually the concurrent maximum value. The overall maximums of all reactions are happening at Load Case 1 with dead load in control. Regarding the magnitude of the maximum reactions, $F_x$ or $F_y$ under Load Cases 1–3 has a relatively small difference between the two structures, but $F_x$ or $F_y$ of the modified structure under Load Cases 4–6 has a considerably larger value than that of the comparison structure. Thus, for the modified structure, the design value of $F_x$ or $F_y$ is controlled by the load cases with participating wind loads, while that of the comparison structure is controlled by the dead load and the live load. This shows that the spiral configuration of the modified structure converts a more significant vertical wind suction or wind pressure to the horizontal plane (X/Y direction) and dissipates it to supports, so it has a better resistance to wind suction or wind pressure than the comparison structure.

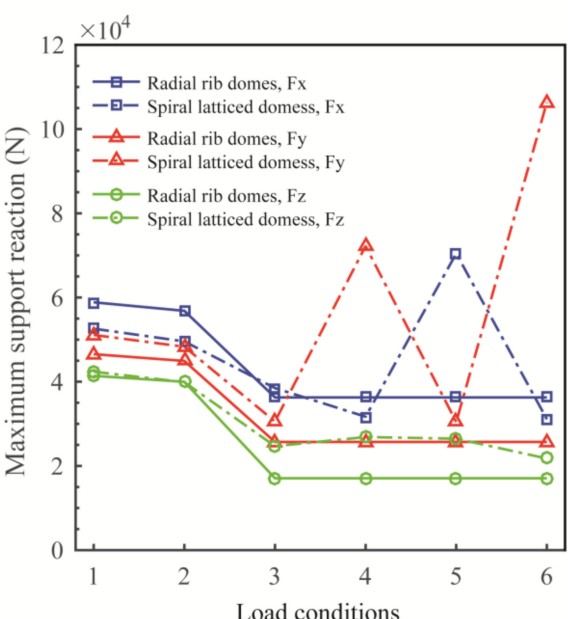

**Figure 5.** Maximum support reactions of the modified structure and the comparison structure under different load cases/conditions.

### 3.2.2. Structural Displacement

Structural displacement reflects the mechanical performance of the structure and affects the serviceability of the structure. Therefore, this section presents the displacement calculation of the modified structure and the comparison structure under different load combinations, as shown in Figure 6. Figure 6 shows that the magnitude of the maximum displacement of both structures under Load Cases 1 and 2 is smaller than that under Load Cases 3–6. Therefore, we focus on the structural displacement under Load Cases 3–6. Due to symmetry, the maximum displacements of the comparison structure are the same under each load case. In contrast, the corresponding maximum displacements of the modified structure are different from each other. The reason is that the comparison group has a symmetrical structure, and its maximum displacement is not affected by the difference in wind direction. On the contrary, the modified structure itself is an asymmetric spiral structure, so its maximum displacement is related to the wind direction angle in the load combination.

Figure 6 also shows that the maximum displacement of the comparison structure is 0.064294 m (load combination control of Case 3–6 with wind load participation). In contrast, the maximum displacement of the modified structure is only 0.043627 m (Case 3 with a wind direction angle of $0°$). Correspondingly, we can easily calculate that the maximum displacement of the modified structure is 32.052% lower than the maximum displacement of the comparison structure. Therefore, the modified structure has better resistance to wind load deformation. Of course, we notice that the maximum displacement of the reference structure and the modified structure meets the design requirements of the deflection limit in the *Technical specification for space frame structures* [14], and the maximum deflection value is less than $L/400$.

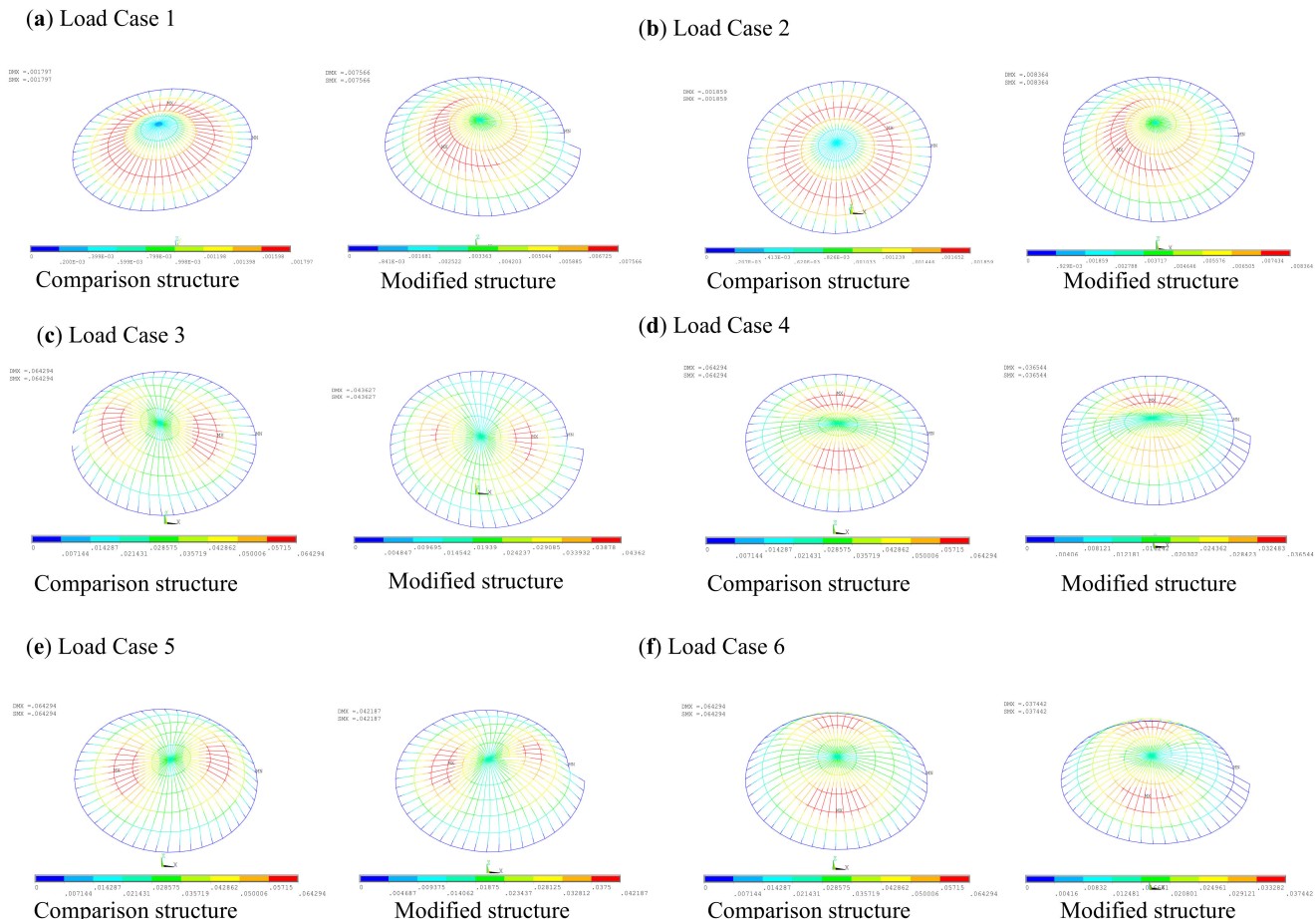

**Figure 6.** Displacement diagrams of the modified structure and the comparison structure under different load cases.

### 3.2.3. Equivalent Stress

Equivalent stress is an important index for characterizing the strength and failure of plastic materials. It follows the fourth strength theory (von Mises theory) and can reflect the mechanical properties of the structure. Therefore, this section calculates the equivalent stress of the comparison structure and the modified structure under different load cases, as shown in Figure 7.

Figure 7 shows that the maximum equivalent stress value of the comparison structure and the modified structure is mainly controlled by the load cases in which wind loads participate. The maximum equivalent stress of the comparison structure has nothing to do with the wind direction angle, and it is always 65.5 MPa under the load combination of Load Cases 3–6. However, the maximum equivalent stress of the modified structure is related to the wind direction angle, which occurs on the windward side, and the range is [48.1 MPa, 66.5 MPa]. In most wind direction angles (Load Cases 3, 4, 6), its equivalent stress is less than that of the comparison structure. At 180 degrees (Load Case 5), its equivalent stress is close to the comparison group. Overall, the equivalent force performance of the modified structure is better than that of the comparison structure. We noted that the maximum equivalent stress of the comparison structure and the modified structure meets the design requirements for the strength of Q235 steel in the *Technical specification for space frame structures* [14].

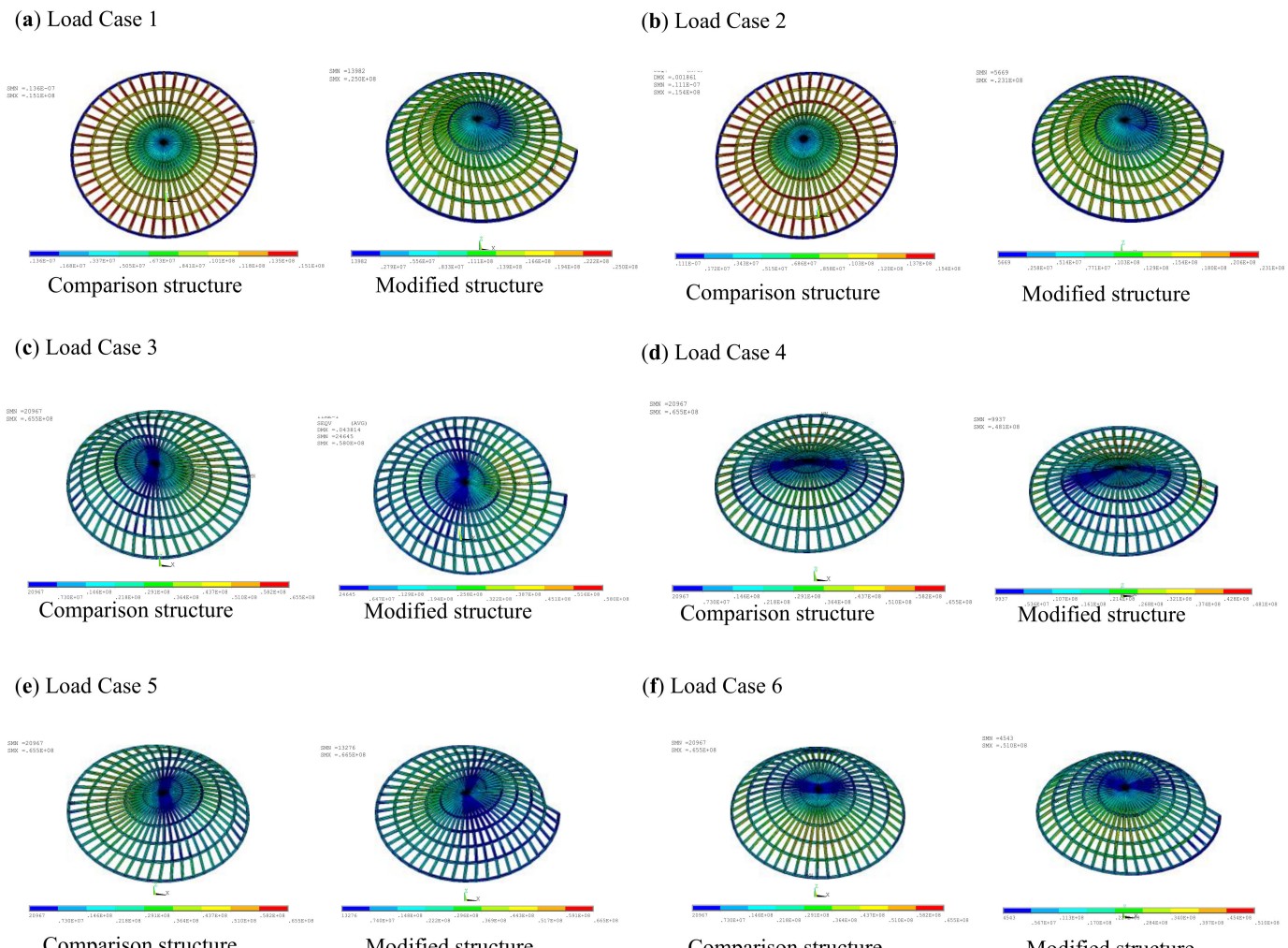

**Figure 7.** Equivalent stress of the modified structure and the comparison structure under different load cases.

## 4. Structural Imperfection and Damage Tolerance Analysis

The modified structure is designed above, and its good performance is preliminarily verified. To better analyze the tolerance of this structure to damage and imperfections, we conduct a comparative study on the modified structure and its comparison structure.

### 4.1. Description of the Calculation Example

The geometric model of the modified structure for each calculation example is shown in Table 3. Its span is 5 m (30 m). *a* is 0.5 m (3 m), *b* is 0.05/$\pi$ m/rad (1.2/$\pi$ m/rad), the ring division frequency is 8 (48), and the number of loops is 5. Two sizes are adopted here to investigate the size effect for completeness. The node coordinates are determined by Equations (1) and (2). The component size is $\phi 60 \times 5.5$ mm ($\phi 219 \times 6$ mm). Other model inputs such as material, element type, or boundary conditions are the same as in Section 3.1. The load is uniform pressure on the external envelope surface (the sphere on which the elements lie) of reticulated shells in the analysis.

The corresponding comparison structures are established as well with the same set of parameters, following the manipulations presented in Sections 2 and 3.1.

**Table 3.** Model parameters of the modified structure and the comparison structure.

| Example Group No. | 1 | | | 2 | | | 3 | | |
|---|---|---|---|---|---|---|---|---|---|
| $r_s$ | 1/5 | 1/6 | 1/7 | 1/10 | 1/15 | 1/20 | 1/5 | 1/6 | 1/7 |
| $r_{\max}$ (m) | 2.5 | 2.5 | 2.5 | 2.5 | 2.5 | 2.5 | 15 | 15 | 15 |
| $f$ (m) | 1 | 5/6 | 5/7 | 1/2 | 1/3 | 1/4 | 6 | 5 | 30/7 |

### 4.2. Tolerance to Node Imperfection

The eigenvalue analysis of the modified structure and the comparison structure without imperfection is shown in Figure 8. The figure shows that the modified structure's eigenvalue is greater than that of the comparison structure, and the deformation rule is the same (antisymmetric or symmetric).

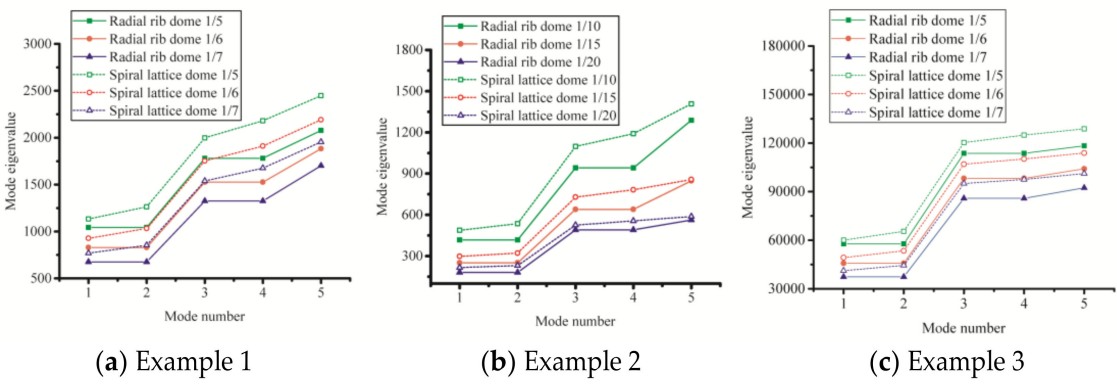

|  (**a**) Example 1 | (**b**) Example 2 | (**c**) Example 3 |

**Figure 8.** The first 5 eigenvalues of the three example groups.

However, when geometric nonlinearity is considered, the relative magnitude of the critical loads of the two ideal structures changes significantly, as shown in Table 4.

**Table 4.** The stable load of the modified structure and the comparison structure in an ideal state.

| Example Group No. | 1 | | | 2 | | | 3 | | |
|---|---|---|---|---|---|---|---|---|---|
| $r_s$ | 1/5 | 1/6 | 1/7 | 1/10 | 1/15 | 1/20 | 1/5 | 1/6 | 1/7 |
| Modified Structure $P_{cr0}$ (kN/m$^2$) | 2.703 | 2.190 | 1.796 | 2.999 | 0.626 | 0.067 | 23.591 | 18.278 | 16.289 |
| Comparison Structure $P_{cr0}$ (kN/m$^2$) | 13.962 | 11.044 | 9.014 | 23.591 | 18.278 | 16.289 | 59.099 | 52.361 | 45.338 |

The reason for this significant change is that the rib-ring type is an axisymmetric shell. Therefore, under ideal conditions, it has spherical membrane tension. Therefore, the critical load of the ideal structure when there is no imperfection is much higher than that of the asymmetric, modified structure without membrane tension.

In practice, the structure cannot be ideal because the node imperfections always exist in reality. Therefore, this section shows the calculations and analysis of the critical load of the modified structure and the comparison structure with joint imperfections. The detailed method to introduce node imperfections and find corresponding stable bearing capacity can be found in our previous research [18]. Here, we consider six initial imperfections with different sizes (L/2000, L/1200, L/1000, L/800, L/500, and L/300, where L is span length) to assess the influence of those imperfections on the stable bearing capacity.

In the analysis, the range of the node imperfection amplitude is [L/2000, L/300]. At the same time, the different $r_s$ of the structure are considered. The related calculation results are shown in Figure 9. "Flat $r_s$" means a flatter shell with smaller rise-to-span ratio $r_s$, compared with "normal $r_s$". For structures with node imperfections, in reality, no

matter what the value of $r_s$ is, the critical load of the modified structure is larger. Hence, its bearing capacity is also greater. The critical load does not always decrease with an increasing imperfection level. When the imperfection is relatively small, the critical load keeps decreasing as the imperfection becomes larger. However, when the imperfection reaches a certain size, the structural configuration changes significantly from the original shape and the structure becomes a new one which might have a higher critical load.

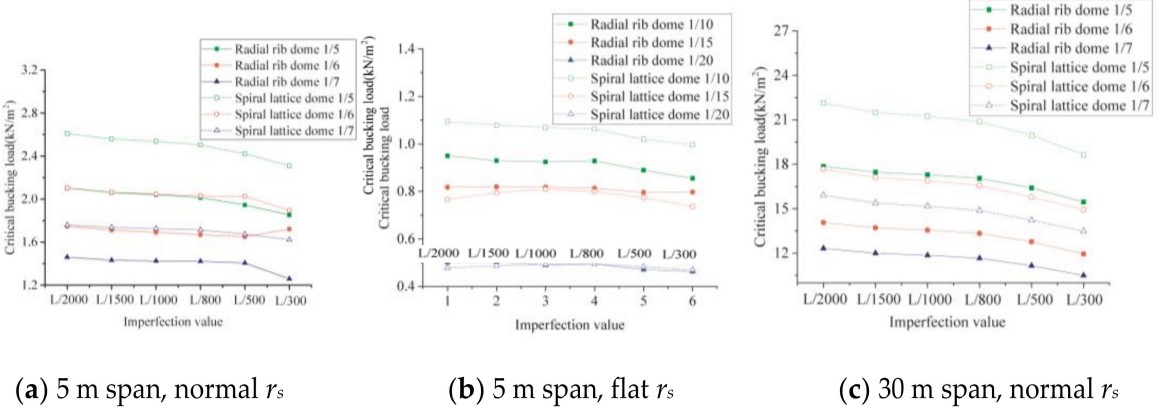

(**a**) 5 m span, normal $r_s$      (**b**) 5 m span, flat $r_s$      (**c**) 30 m span, normal $r_s$

**Figure 9.** $P_{cr}$ of the modified structure and the comparison structure with different $r_s$.

To further analyze the relative difference of the critical load of the modified structure and comparison structure with or without imperfections, we calculate the ratio of critical load when they have imperfections or not, $P_{cr}/P_{cr0}$, and the calculation result is shown in Figure 10. Figure 10 shows that the modified structure is far more resistant to node imperfections than the comparison structure. The primary reason for this high resistance is that the spiral configuration provides a much better load path to mitigate the threats from imperfections.

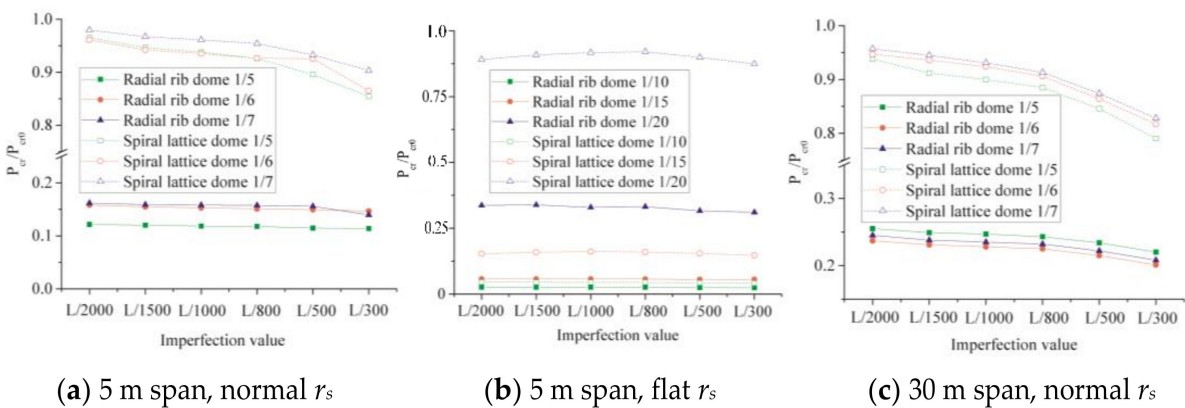

(**a**) 5 m span, normal $r_s$      (**b**) 5 m span, flat $r_s$      (**c**) 30 m span, normal $r_s$

**Figure 10.** $P_{cr}/P_{cr0}$ of the modified structure and the comparison structure with different $r_s$.

*4.3. Tolerance to Damage*

Components are often damaged due to corrosion or manufacturing imperfections. Damage will reduce the bearing capacity of the structure. In order to analyze the extent of the decrease in the bearing capacity of the modified structure and the comparison structure under the same damage degree, we design the following calculation example.

Figure 11 shows the models with component damage for the modified structure and the comparison structure and an enlarged detail of the damaged rods. The geometric span is 30 m and the $r_s$ is 1/5. In the spiral parameters of the geometric model, $a$ is 3 m, $b$ is $1.2/\pi$ m/rad, the ring division frequency is 44, and the number of loops is 5. Its node coordinates are still determined by Equations (1) and (2). The cross-section of beam

elements is $\phi 219 \times 6$ mm. Other model inputs, such as material, element type, or boundary conditions, are the same as in Section 3.1. The corresponding comparison structures can be established as well with the same set of parameters, following the manipulations presented in Sections 2 and 3.1.

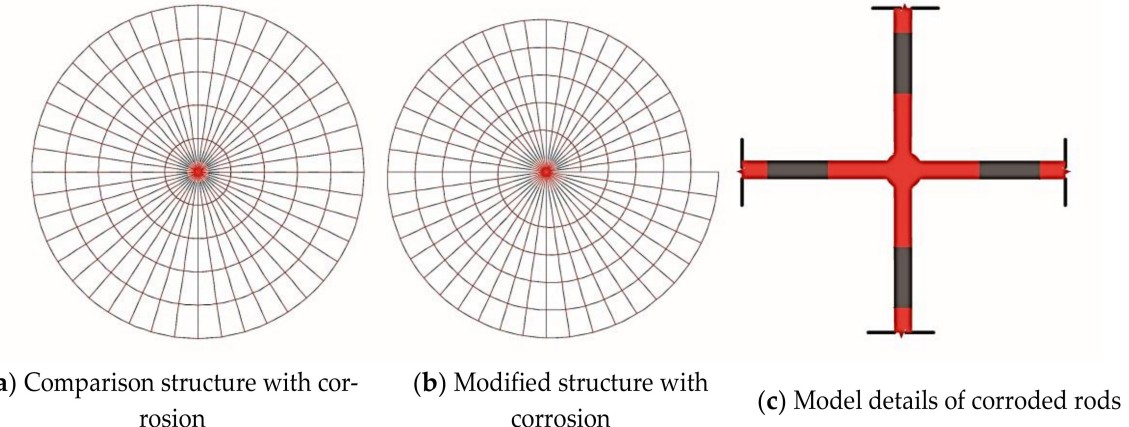

(**a**) Comparison structure with corrosion

(**b**) Modified structure with corrosion

(**c**) Model details of corroded rods

**Figure 11.** Models of the modified structure and the comparison structure with component damage.

In this research, the component damage considered is section loss due to corrosion. The corrosion is assumed to be uniform external corrosion with the same corrosion depth at the middle 1/3 of all rods.

When calculating the critical load values of these two structures, we consider five degrees of component damage (*d*), and define *d* as the ratio of corrosion depth ($h_r$) and pipe thickness ($r_p$), $d = h_r/r_p$. We take five discreet values of *d*: 0.1, 0.15, 0.2, 0.25, and 0.3. The calculation results are shown in Figure 12 below. It can be seen that the greater the damage degree, the lower the critical load of the two structures. Under the same damage degree, the critical load of the modified structure is higher than that of the comparison group. We then calculate the loss of bearing capacity, $D_{cr}$, of the two structures, where $D_{cr}$ = $1 - P_{cr}/P_{cr,0}$, as shown in Figure 13a. In Figure 13, subscripts "r" and "s" mean radial ribbed shell and spiral shell, respectively. The figure shows that the loss of stable bearing capacity of the modified structure is generally lower than that of the comparison shell.

To better measure this tolerance, we propose a modified indicator—tolerance *T* with node imperfections:

$$T_f = P_{cr,0}f / (P_{cr,0} - P_{cr,f}) \qquad (3)$$

*T* with component damage:

$$T_d = P_{cr,0}d / (P_{cr,0} - P_{cr,d}) \qquad (4)$$

*T* with both node imperfections and component damage:

$$T_{f,d} = P_{cr,0}df / \left(P_{cr,0} - P_{cr,d,f}\right) \qquad (5)$$

In the above formulas, $P_{cr,0}$ is the critical load value when the structural damage degree is 0, $P_{cr,f}$ is the critical load value when the structural node imperfection is *f*, and $P_{cr,d}$ is the critical load value when the structural damage degree is *d*. Based on these formulas, the tolerance *T* quantitatively reflects the insensitivity.

We calculate the tolerance of the two structures with both node imperfections and component damage, as shown in Figure 13b. The figure shows that the tolerance of the modified structure is generally higher than that of the comparison structure. The primary reason for this higher tolerance is that the spiral configuration provides a much more reasonable load path to mitigate the threats from structural imperfections and damage. Regarding the huge reduction in the resistance of the comparison structure (traditional

radial ribbed reticulated shell) after the introduction of node imperfections or component damage, many researchers have concluded that it is due to the membrane effect. The radial ribbed reticulated shell can be considered as the skeleton of a membrane shell. The radial and ring members are to resist the radial and hoop stress, respectively. The frame dissipates the load well with ideal and symmetric configuration. However, after introducing imperfections, the radial-ring member assemblies lose the ideal configuration for passing load and are easily snapped. The membrane tension capacity is drastically reduced and leads to a drop in the structural stiffness and bearing capacity. In contrast, the modified structure does not have this issue of high sensitivity to imperfections due to the spiral configuration. The hoop members link to produce a spring-like structure, which causes a spring effect compensating the membrane effect. Thus, the bearing capacity of the spiral reticulated shell will not reduce greatly after introducing imperfections or damage.

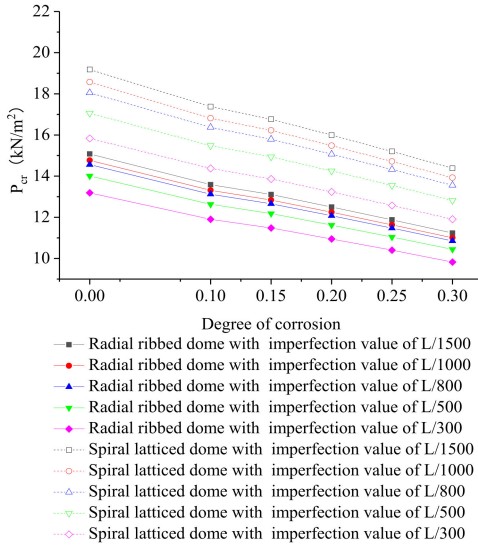

**Figure 12.** $P_{cr}$ of the modified structure and the comparison structure with different damage degrees.

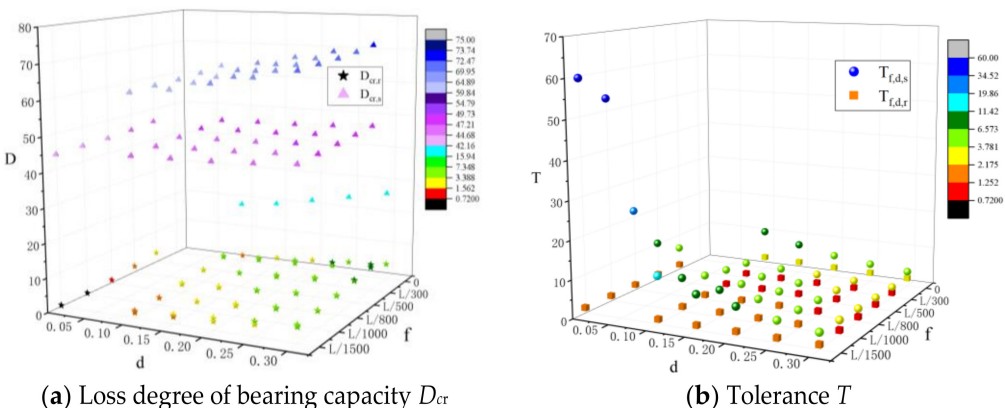

(**a**) Loss degree of bearing capacity $D_{cr}$          (**b**) Tolerance $T$

**Figure 13.** $D_{cr}$ and $T$ of the modified structure and the comparison structure.

## 5. Formulation of the Stable Bearing Capacity of the Modified Structure

Above, we verify that the modified structure has better resistance to imperfections and damage by comparing the traditional structure. However, from the point of view of the load path, the redundancy of the modified structure is not large, and the nonlinearity is strong. Therefore, once the instability failure of brittle characteristics occurs, the consequences will be very unfavorable. Consequently, we must do more virtual calculations on the stable bearing capacity of this structure to develop a stable bearing capacity formula to ensure

that engineers can predict the safety indicators of this type of structure effectively and confidently, in the conceptual design phase.

*5.1. Algorithm*

The method of calculating stable bearing capacity used in this study is a stochastic imperfection modal method based on the response surface that can accurately predict the critical load. It is based on the stochastic imperfection modal superposition method (SIMSM) [18]. However, the difference is that the sampling method in the original method, the Monte Carlo method, is modified to the response surface method. In this way, this modified method based on the SIMSM can consider the randomness of node defects. On the other hand, it adopts the sampling method of the response surface method, which can ensure that the modified method does not require a large amount of sampling and calculation and can be fitted by polynomial functions to obtain ideal analysis results efficiently. The random imperfection expression in the modified method is

$$\{\Delta X\}' = \sum_{i=1}^{m} (r_i\{U_i\}) \tag{6}$$

where $m$ is the mode participation order; $r_1, r_2,..., r_m$ are the participation coefficients, which are independent random variables; and $\{U_i\}$ is the *i-th* order linear buckling mode of the structure, and satisfies

$$\max(U_{i1}, U_{i2}, \cdots U_{in}) = 1 \tag{7}$$

where $n$ is the number of structural joints and $U_{in}$ is the displacement vector of the $n$th node in the $i$th order buckling mode.

Geometric nonlinearity is considered to calculate the stable bearing capacity. The material nonlinearity is not considered since the reticulated shell in this study buckles in the elastic regime. The function fitting of the response surface method of the modified method adopts a polynomial. In other words, the approximate expression of $\hat{p}_{cr}(\underline{r})$ is now polynomial. If the quadratic complete polynomial is taken, then

$$\hat{p}_{cr}(\underline{r}) = a_0 + \sum_{i=1}^{n} a_i r_i + \sum_{i=1}^{n} \sum_{j=i}^{n} a_{ij} r_i r_j \tag{8}$$

where, $a_0$, $a_i$, and $a_{ij}$ are the undetermined coefficients.

$P_{cr}$ is tested to show a normal distribution $N\left(\mu_{P_{cr}}, \sigma_{P_{cr}}^2\right)$. Therefore, following the $3\sigma$ rule of the normal distribution, the most unfavorable critical load $p*$ by the modified method is

$$p* = \mu_{p_{cr}} - 3\sigma_{p_{cr}} \tag{9}$$

Other detailed theoretical derivations and verifications of this new method will be discussed in another paper to be published soon. This new method can find more unfavorable critical loads than the traditional method, as shown in Figure 14. In this study, when using this efficient random defect mode method, the combined buckling mode selects the first three orders. Its participation coefficients are independent of each other and follow a uniform distribution $[-1, 1]$, and the range of sampling times is $[10, 5000]$.

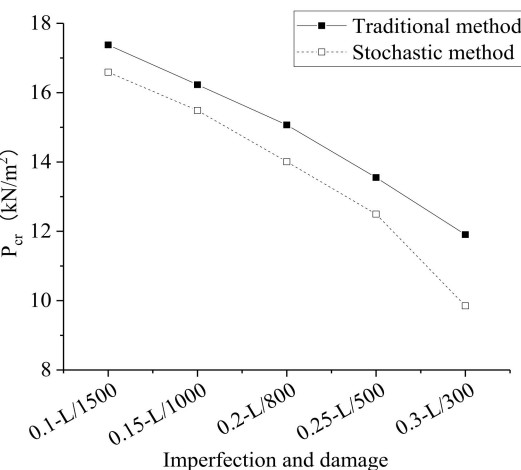

**Figure 14.** Critical loads detected by traditional and stochastic method regarding the imperfection and damage degree.

### 5.2. Numerical Models

The finite element model of the modified structure used in this section is the same as that in Section 3. Its imperfection amplitude is L/1000, and its damage degree is 0.1. The calculation result of sensitivity analysis is shown in Figure 15. It can be seen from the figure that when sampling 2000 times, the critical load of the modified structure reaches a reliable convergence value.

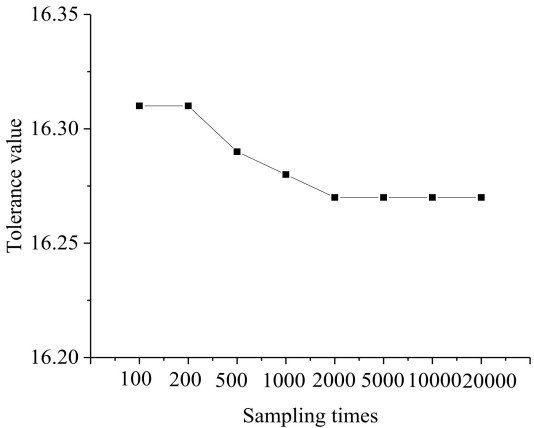

**Figure 15.** Convergence process of tolerance value of the modified structure.

### 5.3. Formula for Stable Bearing Capacity of the Modified Structure

It is easy for us to repeat the above steps and go through a lot of parameter analyses on the modified structure. In this way, we can obtain the critical loads of the structure with different parameters. Then, we use the continuous analogy method [19] to fit them mathematically. In this way, we can obtain its critical load calculation formula, as shown in Equation (10).

$$q_{cr} = 0.2\sqrt{BD}/r^2 + 1.18 \tag{10}$$

In the formula, $B = (B_{\phi\phi} + B_{\theta\theta})/2$, $D = (D_{\phi\phi} + D_{\theta\theta})/2$, $B_{\phi\phi} = E\delta_{\phi}$, $D_{\phi\phi} = D_{\phi}$, $B_{\theta\theta} = E\delta_{\theta}$, $D_{\theta\theta} = D_{\theta}$, $B_{\phi\theta} = B_{\theta\phi} = 0$, $D_{\phi\theta} = D_{\theta\phi} = 0$, $B$ is the film stiffness, $D$ is the bending stiffness, $r$ is the radius of curvature of the spherical shell, $q_{cr}$ is the stable bearing capacity, $E$ is the elastic modulus of the rod, and $\delta_{\phi}$, $\delta_{\theta}$, $D_{\phi}$, $D_{\theta}$ are the equivalent thickness and flexural rigidity. The detailed calculating procedures for all these parameters follow the reference [18] and are not presented here for conciseness. Equation (10) has a simple form and gives effective results of structural stable bearing capability. Thus, it can help

engineers to find out the safety indicators of the modified structure quickly during the conceptual design phase.

## 6. Conclusions

This study proposes a modified and bionic structure—a spiral single-layer reticulated shell—to be used for building structures. According to the current Chinese design codes, its mathematical model and geometric model are rationally designed, and calculation examples are used to analyze its tolerance to damage and imperfections. Finally, the thin-shell pseudo-shell method is used to establish a unified bearing capacity formula. Based on demonstrations and discussions, we obtain some valuable regularity findings:

(1) The spiral structure is more stable than the traditional structure and has better corrosion and damage tolerance.
(2) The spiral line of the new design can successfully convert the vertical wind pressure (wind suction) into the plane reaction force of the structural support, therefore, it has better wind resistance.
(3) The unified stable bearing capacity formula of the modified structure allows engineers to consider its structural performance when choosing a model with high accuracy and to thus make more effective model selection decisions.

**Author Contributions:** Conceptualization, H.L.; methodology, H.L., F.L., H.Y., D.A., and C.X.; software, F.L.; formal analysis, H.L., F.L., H.Y.; investigation, C.X.; resources, H.L.; data curation, H.L.; writing—original draft preparation, H.L., F.L., and H.Y.; writing—review and editing, H.L., H.Y., D.A., and C.X.; visualization, F.L.; supervision, H.L.; project administration, H.L.; funding acquisition, H.L. All authors have read and agreed to the published version of the manuscript.

**Funding:** This research was funded by National Natural Science Foundation of China (grant number 51708135, 52068003), the Guangxi Natural Science Foundation (grant number 2019GXNSFAA185057), the Opening Project of Guangxi Laboratory on the Study of Coral Reefs in the South China Sea (grant number GXLSCRSCS201900*), and the Opening Project of Guangxi Key Laboratory of Disaster Prevention and Engineering Safety (grant number 2019ZDK043, 2019ZDK044). Opinions, findings, and conclusions expressed in this paper are those of the authors and not necessarily those of the sponsors.

**Institutional Review Board Statement:** Not applicable.

**Informed Consent Statement:** Not applicable.

**Data Availability Statement:** The raw data required to reproduce these findings cannot be shared at this time due to time limitations. The processed data required to reproduce these findings cannot be shared at this time due to time limitations.

**Acknowledgments:** This work was supported by the National Natural Science Foundation of China (grant number 51708135, 52068003), the Guangxi Natural Science Foundation (grant number 2019GXNSFAA185057), the Opening Project of Guangxi Laboratory on the Study of Coral Reefs in the South China Sea (grant number GXLSCRSCS201900*), and the Opening Project of Guangxi Key Laboratory of Disaster Prevention and Engineering Safety (grant number 2019ZDK043, 2019ZDK044). Opinions, findings, and conclusions expressed in this paper are those of the authors and not necessarily those of the sponsors. The authors are particularly grateful to W.C.X. Ample. He is a freelancer and American citizen of Chinese descent, and provided valuable suggestions on the structure and language used in this paper.

**Conflicts of Interest:** The authors declare no conflict of interest.

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
