# Peer review of "A Spiral Single-Layer Reticulated Shell Structure: Imperfection and Damage Tolerance Analysis and Stability Capacity Formulation for Conceptual Design"

_buildings, doi:10.3390/buildings11070280_

Round 1

Reviewer 1 Report

The article is interesting and raises interesting issues. It is also linguistically well written, but requires some changes to make it even more understandable. 

  1. The number of articles cited is far too high. Personally, I would cut it in half, but I understand that it will be difficult for authors to decide on such dresting steps. Many articles are very old, but I don't mind that much, although unfortunately it indicates that the problem presented is a bit forgotten (although still interesting to me). It is definitely an exaggeration to cite 6 articles about geometric imperfection in one place (5-10, line 29). With this way of citing, there might as well be 60 of them.
  2. Line 80: explain precisely rise-to-span ratio (rs). The lack of a precise definition makes it difficult to understand the rest of the article. 
  3. Lines 99, 199 and 255: elastic modulus is 201 GPa and shear modulus is 1096 MPa? Here is some misunderstanding. For steel, if E = 201 GPa, then the shear modulus should be G = E / 2 (1 + ni), which is about 80 GPa. This must be clarified.
  4. Subsection 3.2.1 should be completely revised due to imprecise (and inadequate) wording. 
    It is difficult for me to agree with the statement that the structure "generates" reactions. 
    In the sentence "The figure shows..." (lines 124-125) improve grammar.
    The sentence "At the same time ..." (lines 126-127) is absolutely unclear.
    The expression "...reaction force is controlled by the combination of ... loads" (lines 131-132) is unacceptable to me. The word "controlled" is incorrect. 
  5. The sentence "For example, the reaction force of each support of the new structure has larger difference compared with the compassion structure." (lines 128-129) shows the imperfection of subsection 3.2.1. The authors write about the reaction, but the support boundary conditions have not yet been defined! The authors write about a comparison structure (not "compassion"), but the description of this structure is very vague. 
  6. Line 142: "Therefore, this section calculates...". Section does not calculate.
  7. Figure 6 (line 165): there are no Figure 6.
  8. Line 167: "Equivalent stress" What kind of equivalent stress (Mises)? There are a lot of hypotheses which can be used to calculate an eqivalent stress.
  9. Line 200: shouldn't be "188 beam elements"? Line 201 "fixed hinge support" What does it mean? Shouldn't it be "hinge support"?
  10. Lines 202-203: "The comparison structure can be established according to the principles in Section 2 according to the model parameters of the new structure." Based on Section 2, it is not possible to define what the comparative construction looks like. 
  11. Lines 276-281: improve editorial.
  12. Line 330: explain why equation (9) gives the most unfavorable critical load.
  13. Line 338: Figure 14 in my opinion is wrongly named. This is not "Verification of effectiveness". The name "Imprfection and damage" is also unclear.
    I also wonder if the connection of the points on the graph is correct. What parameters of imperefection and damage are between the points? 
  14. Coclusions, line 362: "This paper proposes a new bionic structure..." I think that's an exaggeration. In the context of cited papers [20-27], this article does not propose a new structure. 
  15. As the comparative structure was not defined (even the weight was not compared, the cross-sections of the elements were not given, etc.), it is very difficult to refer to the presented results. In particular, it is surprising that there is no attempt to explain the huge reduction in the resistance of the comparative structure after the introduction of imperfections. The explanation that this is due to symmetry is somewhat vague. 

Reviewer 2 Report

- General comment

In this article the authors present a proposal for a spiral single-layer reticulated shell structures based on the spider web configuration. In order to demonstrate the viability of this new configuration, they present a comparative analysis with an equivalent structure with the usual configuration. This analysis addresses, among other aspects, stability and tolerance to damage and imperfections. The authors also propose a formula to help engineers choose the most efficient solution.

In my opinion the theme seems to be interesting and with the possibility of practical application.

The article is well structured, but needs to be deepened/developed in some sections. Some figures are of poor quality and others are lacking. Some references appear to be incorrect.

I recommend that the authors review the article and resubmit it as a new article, mainly because it refers to an unpublished study and the changes to implement seem to me to be quite a lot. However, I made some comments to improve the quality of the future article.  I recommend that the authors take the observations into account and submit the new article.

- Specified comments

Comment 01: page 1, line 30-31

I think that the division of the word "capacity” is not correct. During the text there are numerous situations similar to this one. Some words are well divided, but it seems to me that others are not. Please check.

My suggestion is that you should avoid splitting words.

Comment 02: page 1, line 59

It seems to me that references 15 to 19, 21 and 23 to 27 have nothing to do with the text where they are mentioned. You must check these situations and if necessary correct or change.

Comment 03: page 2, line 49-51

Just an example of a situation similar to comment 1. You should check the entire text.

Comment 04: page 3, Figure 2

In Figure 2 it indicates that the height is 5 m, so in the legend it should put that the figure is for the case where rs is 1/6.

Comment 05: page 3, Figure 3

Similar to comment 4.

Comment 06: page 3, line 101

If you use Ansys you must put at least one reference to the software in the References.

Comment 07: page 5, Figure 5

In my view, the Figure has poor quality, it should be improved to be more readable.

Comment 08: page 5, Figure 6

Figure 6 is missing.

Comment 09: page 6, Figure 7

In my view, the Figure has poor quality, it should be improved to be more readable. You must standardize the descriptions and not put them on top of the captions. Instead of having a legend for each graph, you can keep the same legend for all graphs and place it larger and beside or below all graphs so that the values can be seen.

Comment 10: page 6, Section 4

It seems to me that you used a reduced model for the calculations. Why did you do it? What are the criteria for its design? It seems to me that this makes the analysis more confusing.

You must state in the text the reasons for your choices.

Comment 11: page 7, Table 3

The formatting of the table seems a little confusing to me, you should reword it. The same can be improved on the others. It should also clarify what is in the table: Examples, S, etc.

Comment 12: page 7, Figure 8

Similar to comment 7. The font size is very small, you must increase it to be able to read better.

Comment 13: page 7, Table 4

Similar to comment 12.

What is f/L? Is it the S in Table 3? You should clarify this situation and similar ones that appear throughout the article.

Comment 14: page 8, Figure 9

Similar to comment 12.

What is "normal rs" and "flat rs"? You must indicate what these designations mean.

Comment 15: page 8, Figure 10

Similar to comment 12.

Comment 16: page 9, Figure 11

The figure is not very explicit, it is not clear what it wants to represent.

Comment 17: page 9, line 273

I think instead of "superscript" it should be "subscript", but in the figure it's not noticeable (low quality).

Comment 18: page 10, Figure 12 and 13

Similar to comment 12.

Comment 19: page 11, Equation (9)

It should put what the meaning of the terms of the equation.

Comment 20: page 11, line 331 to 338

You cannot present results based on a method that you intend to present in a future article. Either publish the article first, or present here a brief description of the method and its application, or do not list the results obtained.

Comment 21: page 12, line 354 to 357

Problems with formatting. Please check.

It should better explain what the terms are, what they mean and how they are obtained.

Comment 22: page 12, Before "6. Conclusions"

For a more effective connection to the project, it is necessary to present an explicit example. It can be an application to one of the models presented above. With all the steps that the engineer must take.

Allow me to make a suggestion now, given that in the spiral structure all supports have different elevations, why not consider an additional turn to the same elevation, the supports being identical to the traditional situation and of a more current practical application.

Comment 23: page 13, line 415

There is a blank line. You must delete it.

Comment 24: page 14, line 436

It must keep the same formatting used in the previous references, ie, DONG ... -> Dong ...

Please check.

Reviewer 3 Report

Comments and Suggestions for Authors there are in the attached file (peer-review-12280069.v1). 

Reviewer 4 Report

see attached file "comments_authors.pdf"

Round 2

Reviewer 1 Report

The authors of the article tried to consistently introduce comments from reviewers. I believe the article looks pretty good now, and may be published in a journal with minor editorial corrections. 

Reviewer 2 Report

After the changes introduced by the authors, it seems to me that the article is better.

Therefore, I recommend that the article be accepted for publication.